# Supplementary Far-Red and Blue Lights Influence the Biomass and Phytochemical Profiles of Two Lettuce Cultivars in Plant Factory

**DOI:** 10.3390/molecules26237405

**Published:** 2021-12-06

**Authors:** Yamin Li, Linyuan Wu, Haozhao Jiang, Rui He, Shiwei Song, Wei Su, Houcheng Liu

**Affiliations:** College of Horticulture, South China Agricultural University, Guangzhou 510642, China; yaminli@stu.scau.edu.cn (Y.L.); linyuanwu@stu.scau.edu.cn (L.W.); jhzh111@stu.scau.edu.cn (H.J.); ruihe@stu.scau.edu.cn (R.H.); swsong@scau.edu.cn (S.S.); susan_l@scau.edu.cn (W.S.)

**Keywords:** biomass, coloration, lettuce, light-emitting diodes, phytochemicals, plant factory, vitamins

## Abstract

Three different LED spectra (W: White light; WFR: W + far-red light; WB: W + blue light) with similar photosynthetic photon flux density (PPFD) were designed to explore the effects of supplementary far-red and blue lights on leaf color, biomass and phytochemicals of two cultivars of red-leaf lettuce (“Yanzhi” and “Red Butter”) in an artificial lighting plant factory. Lettuce plants under WB had redder leaf color and significantly higher contents of pigments, such as chlorophyll a, chlorophyll b, chlorophyll (a + b) and anthocyanins. The accumulation of health-promoting compounds, such as vitamin C, vitamin A, total phenolic compounds, total flavonoids and anthocyanins in the two lettuce cultivars were obviously enhanced by WB. Lettuce under WFR showed remarkable increase in fresh weight and dry weight; meanwhile, significant decreases of pigments, total phenolic compounds, total flavonoids and vitamin C were found. Thus, in the plant factory system, the application of WB can improve the coloration and quality of red leaf lettuce while WFR was encouraged for the purpose of elevating the yield of lettuce.

## 1. Introduction

Indoor farms, especially plant factories with artificial light (PFALs), can produce high-value crops and increase crop production in a stable, controllable and safer manner [1]. Lettuce (*Lactuca sativa* L.) is one of the most common vegetables in PFALs because it is suitable for hydroponic growth and has a short growth period. Lettuce is popular as a salad vegetable, not only due to its tender taste, sweet and refreshing flavor and ornamental morphology with vivid colors, but also the abundant nutritional and antioxidant compounds such as vitamin C, sugars, proteins and phenolics [2].

Different light conditions (light quality, light intensity and photoperiod) evoke different morphogenetic and photosynthetic responses in plants and the responses vary among plant species [3,4,5]. Blue lights (400–500 nm), included in the photosynthetically active radiation (PAR), can be predominantly absorbed by chlorophylls and function as the major energy sources in plants photosynthesis [3]. Blue light also strongly impacts the morphology and nutritional qualities in plants. Blue light inhibits the hypocotyls elongation and leads to dwarfed plants. The hypocotyl length and cotyledon area in kale and mustard had negative correlations with the proportion of blue light (5–30%) in red and blue LED lighting [6]. However, canopy height, stem length and internode length of petunia, calibrachoa, geranium and marigold were markedly improved by blue light [7], and leaf angle of these four bedding plants were significantly enlarged by red light. Compared to white light, the cucumber leaves accumulated more N, K, Ca, Mg and Mn under monochromatic blue light [8]. With respect to phytochemicals, blue light positively induced the lycopene content in tomato fruit, contents of anthocyanins, chlorophylls and carotenoids in tomato leaf and anthocyanins content in tomato seedlings [9]. Moreover, wide spectra lighting with relatively high blue and green light percentage induced the highest total phenolic content in pomegranate [10].

Far-red light (beyond 700 nm) has long been considered ineffective in photosynthesis for its poor absorption by mesophyll cells. It is known to induce shade-avoidance syndrome in plants, with the traits of increased plant height, reduced stem diameter and leaf area, narrower leaf angle and accelerated flowering [11,12]. Far-red light might lead to the reduction in yields and weakened immunity to pests and pathogens in field cultivation [13,14,15]. Lettuce grown with supplemental far-red light exhibited the lowest biomass [16], which was probably associated with decreased contents of chlorophyll and carotenoid. In other aspect, far-red light could promote the photosynthetic efficiency via increasing the quantum yield of photosystem II, attenuating the non-photochemical quenching of fluorescence and reducing the dissipation as heat [17]. In this regard, even though the contents of xanthophylls, β-carotene and chlorophyll were significantly reduced by additional far-red light, the fresh weight and dry weight of “Red Cross” lettuce was dramatically enhanced [18]. Moreover, far-red light was efficient in increasing the contents of ascorbic acid and soluble sugar in lettuce [16] but reduced the contents of glucose and fructose in tomato fruits [19] and impeded the accumulation of caffeic acid and chicorid acid in *C. denticulatum* plants [20]. Previous studies suggested that far-red light acted more likely as a signal for plant growth, development and metabolism, through mediating the key enzymes that participate in related reactions or metabolic pathways [16,17,19,20].

As mentioned above, different lights and their combinations drive differential changes in plant architecture, photosynthesis and biosynthesis of phytochemicals. The light source is entirely artificial light in PFAL systems, which allows precise studies of effects of a particular light quality on plants by changing a narrow band of spectra. Therefore, in this study, the equalized-white LED lights whose band was basically consistent with PAR were used as control, the changes of leaf coloration, biomass and phytochemicals of two cultivars of red leaf lettuce in response to supplementary far-red light and blue light under similar PPFD were detected. Meanwhile, this study applied the heat map analysis and principal component analysis to achieve overall data visualization and simple clustering, which helped to fully evaluate the effects of supplementary light on lettuce. This study aimed to provide reference for the efficient LED light model with high productivity in PFAL systems.

## 2. Materials and Methods

### 2.1. Plant Material and Growth Conditions

This experiment was carried out in the PFAL, South China Agriculture University (east longitude 113.36°, north latitude 23.16°). The PFAL was equipped with a six-layer vertical hydroponic system using the deep flow technique (DFT, liquid level at 7 cm each layer). Each layer accommodated 15 planting plate and was divided into five cultivation regions whose light spectra can be adjusted independently. Seeds of red oakleaf lettuce (*Lactuca sativa* cv. “Yanzhi”, ShouheSeed, China) and red butterhead lettuce (*Lactuca sativa* cv. “Red Butter”, ShouheSeed, China) were soaked for 1 h, then sowed in a moist sponge block (2 cm × 2 cm × 2 cm) and kept in the dark germination chamber. After 72 h, the germinated seeds with sponge blocks were grown in the DFT with half-strength nutrient solution. The full-strength nutrient solutions (EC ≈ 1.2 mS·cm^−1^ and pH ≈ 6.4) were composed of the following elements: 56.0 mg·L^−1^ N, 22.8 mg·L^−1^ P, 184.7 mg·L^−1^ K, 80.0 mg·L^−1^ Ca, 24.0 mg·L^−1^ Mg, 64 mg·L^−1^ S, 2.8 mg·L^−1^ Fe, 0.5 mg·L^−1^ B, 0.5 mg·L^−1^ Mn, 0.05 mg·L^−1^ Zn, 0.02 mg·L^−1^ Cu, 0.01 mg·L^−1^ Mo, temperature 21 ± 2 ℃, CO_2_ concentration 400–600 μmol·mol^−1^, relative humidity 55–60%, and equalized-white LED lighting at 250 μmol·m^−2^·s^−1^ PPFD from 8:00 to 18:00. After 18 days, the seedlings with three expanded true leaves were transplanted into the planting plate (90 cm × 60 cm, 24 plants per plate) with full-strength nutrient solution.

### 2.2. Light Treatments and Sample Collection

The adjustable LED panels (Chenghui Equipment Co., Ltd., Guangzhou, China; 150 cm × 30 cm) with equalized-white (peaking at 440 nm and 660 nm), far-red (730 ± 10 nm) and blue (450 ± 10 nm) LEDs were used as light sources. LED lightings at PPFD of 250 μmol·m^−2^·s^−1^ (10 h·day^−1^ from 8:00 to 18:00). There were three light regimes (Table 1): W (equalized-white light), WFR (W + far-red light) and WB (W + blue light). Twenty days after treatments, two cultivars of lettuce were sampled randomly with four biological repetitions (12 plants per repetition). The pigment contents were detected immediately after sampling, while samples for biochemicals and antioxidant activity assays were collected by liquid nitrogen and kept at −70 ℃.

### 2.3. Color Measurement

The foliage color of lettuce was measured with a colorimeter (CR-10 plus, Konica Minolta Inc, Tokyo, Japan). The notation L* means lightness; a* represents the color from green to red; b* suggests the color from blue to yellow.

### 2.4. Chlorophyll and Carotenoids Measurements

Fresh tissue (0.5 g) from the mature leaves of lettuce was extracted with acetone-ethanol (1:1, *v*/*v*) solution and kept in darkness, 4 ℃ for 24 h until the leaf tissue turned white. Then the filtrates were measured at 645 nm, 663 nm and 440 nm by a UV-spectrophotometer (Shimadzu UV-16A, Shimadzu Corporation, Kyoto, Japan). Contents of chlorophyll a (Chl a), chlorophyll b (Chl b), chlorophyll (a + b) (Chl (a + b)) and carotenoids (Car) were calculated according to the following equations [21]:Chl a (mg·g^−1^ FW) = (12.70 × OD_663_ − 2.69 × OD_645_) × *V*/1000 *W*,(1)
Chl b (mg·g^−1^ FW) = (22.88 × OD_645_ − 4.67 × OD_663_) × *V*/1000 *W*,(2)
Chl (a + b) (mg·g^−1^ FW) = (8.02 × OD_663_ + 20.20 × OD_645_) × *V*/1000 *W*,(3)
Car (mg·g^−1^ FW) = (4.70 × OD_440_ − 2.17 × OD_663_ − 5.45 × OD_645_) × *V*/1000 *W*(4)
where *V* is the volume of extract solution (mL), and *W* is the fresh weight (g) of the sample.

### 2.5. Total Anthocyanins Measurements

The total anthocyanin (TA) measurement was analyzed as previously described by Rapisarda et al. [22]. Two groups of fresh samples (1.0 g) were extracted, respectively, with pH 1.0 buffer solution (50 mM KCl and 150 mM HCl) and pH 4.5 buffer solution (400 mM CH_3_COONa and 240 mM HCl). These extractions were centrifuged at 12,000 rpm, 4 ℃ for 5 min. Then the supernatants were determined at 510 nm by using the UV-spectrophotometer. The TA values were calculated with the following equation:
TA (mg·g^−1^ FW) = (*A*_1_ − *A*_2_) × 484.8 × dilution factor/24.825(5) where *A*_1_ is the absorbance of the sample extracted from pH 1.0 buffer solution, *A*_2_ is the absorbance of the sample extracted from pH 4.5 buffer solution. The number 484.8 represents the molecular weight of cyaniding-3-glucoside chloride. The dilution factor in this measurement is 1. The number 24.825 was the absorption coefficient at 510 nm.

### 2.6. Fresh and Dry Weight Measurements

Nine plants of lettuce in each treatment were randomly selected to measure the fresh and dry weight. The fresh weight (FW) was determined by analytical balance within 5 min upon harvest. Then these samples were oven-dried at 70 ℃ for 60 h to determine the corresponding dry weight (DW).

### 2.7. Nutritional Compounds Measurements

The soluble proteins (SP) were determined according to Blakesley and Boezi [23]. Fresh lettuce samples (1.0 g) were added into 8 mL distilled water. After being centrifuged at 8000 rpm for 10 min, 1 mL supernatant was mixed with 5 mL Coomassie brilliant blue G-250 solution; 5 min later, the absorbance was measured at 595 nm by the UV-spectrophotometer.

The soluble sugars (SS) were analyzed following the method described by Kohyama and Nishinari [24]. Fresh lettuce samples (1.0 g) were twice extracted with 80% ethanol (*v*/*v*) and activated carbon powder (10 mg), and 80 ℃ water bath for 40 min each time. The mixture was filtered and then diluted to a total volume of 25 mL with 80% ethanol. Then 0.2 mL extract was mixed with 0.8 mL diluted water and 5 mL sulfuric acid anthrone reagent and 100 ℃ water bath for 10 min. The absorbance was detected at 625 nm by the UV-spectrophotometer.

The nitrates were determined with the method proposed by Cataldo et al. [25]. Fresh lettuce samples (1.0 g) were mixed with 10 mL deionized water and 100 ℃ water bath for 30 min. The extract was filtered and diluted with deionized water to a total volume of 25 mL. Then 0.1 mL filtrate was mixed with 0.4 mL 5% salicylic acid (*w*/*v*, dissolved in H_2_SO_4_) reagent. Ten minutes later, 9.5 mL 8% NaOH (*w*/*v*) was added. The absorbance was measured at 410 nm with the UV-spectrophotometer.

### 2.8. Antioxidant Activity Measurements

Fresh lettuce samples (0.5 g) were soaked in 8 mL methanol for 30 min in darkness. After 15 min centrifugation at 3000 rpm, the supernatant was kept in darkness, 4 ℃ for the measurements of DPPH radical inhibition percentage (DPPH) and ferric ion reducing antioxidant power (FRAP).

The DPPH measurement was based on the method of Musa et al. [26]. Three types of mixture needed to be prepared (*A_i_*: Supernatant of 2 mL mixed with 2 mL 0.2 μM DPPH; *A_j_*: Supernatant of 2 mL mixed with 2 mL ethanol; *A_c_*: 0.2 μM DPPH mixed with 2 mL ethanol) and determined at 517 nm by the UV-spectrophotometer. The DPPH radical inhibition percentage was calculated as following:

DPPH (%) = [1 − (*A_i_*−*A_j_*)/Ac] × 100%(6)

The FRAP measurement was according to Tadolini et al. [27]. The FRAP reagent was prepared by mixing 300 mM acetate buffer (pH 3.6), 20 mM ferric chloride and 10 mM 2,4,6-tripyridyl-S-triazine (TPTZ) in 40 mM HCl in the proportion of 10:1:1 (*v*/*v*/*v*). The supernatant (0.4 mL) was added to the FRAP working reagent (3.7 mL), and the mixture was preserved in a 37 ℃ water bath for 10 min. The absorbance was determined at 593 nm by the UV-spectrophotometer. FeSO_4_·7H_2_O was used as the standard, and the results were expressed as mmol·g^−1^ FW.

### 2.9. Antioxidant Components Measurements

Fresh lettuce samples (0.5 g) were soaked in 8 mL methanol for 30 min in darkness. After 15 min centrifugation at 3000 rpm, the supernatant was kept in darkness, 4 ℃ for the measurements of total phenolic compounds (TPC) and total flavonoids (TF).

The TPC measurement was conducted as stated by Tadolini et al. [27]. The supernatant (1.0 mL) was mixed with 0.5 mL Folin-ciocalteu ultra-pure water reagent (1: 1, *v*/*v*) and 1.5 mL 26.7% Na_2_CO_3_ solution (*w*/*v*). The mixture was diluted to a total volume of 10 mL with ultra-pure water. After 2 h of reaction, the absorbance was recorded at 760 nm with the UV-spectrophotometer. TPC values were calculated from the gallic acid standard curve, and the results were expressed as mg gallic acid equivalent fresh weight (mgGAE·g^−1^ FW).

The TF measurement was in accordance with the method used by Sanchez-Rangel et al. [28]. The supernatant (1 mL) was mixed with 10 mL 30% ethanol (*w*/*v*) and 0.7 mL NaNO_2_ solution (*w*/*v*). After 5 min, 0.7 mL 10% Al(NO_3_)_3_ solution (*w*/*v*) was added in for 5 min’ reaction. Then, 5 mL 5% NaOH solution (*w*%) and 8.6 mL 30% ethanol were added. The absorbance was determined at 510 nm with the UV-spectrophotometer. Rutin hydrate was used as the standard, and the results were expressed as mg·g^−1^ FW.

Vitamin C (VC) content was analyzed according to Li et al. [29]. Fresh lettuce samples (1.0 g) were soaked and diluted to a total volume of 50 mL with EDTA-oxalic acid solution (200 mM EDTA and 50 mM oxalic acid) and centrifuged at 5000 rpm for 5 min. Supernatant of 10 mL was mixed with 1 mL 3% HPO_3_ solution (*w*/*v*), 2 mL 5% H_2_SO_4_ (*v*/*v*) and 4 mL 5% H_8_MoN_2_O_4_ (*v*/*v*). After 15 min, the absorbance was taken at 705 nm by the UV-spectrophotometer.

Vitamin A (VA) contents were determined using the plant vitamin A ELISA Kit (Mlbio, Shanghai, China), according to the manufacturer’s protocol. The absorbance was measured at 450 nm by the microtiter plate reader and the VA contents were recorded by comparing the absorbance to the standard curve.

### 2.10. Heat Map Analysis

The overall data were firstly normalized using the method of Z-score and centered to the median of the entire sample set to lower the relatively large differences in different parameters. Then, the transformed data were visualized into heat map by TBtools software (v1.09867) using the function called Heatmap Illustrator [30].

### 2.11. Statistics Analysis

Data were analyzed by a one-way analysis of variance (ANOVA) and two-way analysis of variance, SPSS software (v25.0, SPSS Inc., Chicago, IL, USA) with Tukey’s honest significant difference tests. All values were reported as the means of four replicates with standard deviations (SD). The principal component analysis was performed with GraphPad Prism 9 (v9.1, GraphPad Software, San Diego, CA, USA, www.graphpad.com).

## 3. Results

### 3.1. Plant Growth and Biomass

The growth and biomass of lettuce plant were significantly affected by light quality (*p* < 0.001) (Figure 1 and Table 2). Under the basic W light, two lettuce cultivars showed almost the same fresh weight and dry weight (Table 2). Both “Yanzhi” and “Red Butter” obtained significant higher plant fresh weight (45.91, 93.11%), shoot fresh weight (48.10, 97.34%), plant dry weight (43.66, 53.57%) and shoot dry weight (51.44, 53.57%) under WFR (Table 2). However, “Red Butter” exhibited significant lower plant fresh weight (17.16%), shoot fresh weight (18.17%), root fresh weight (6.16%), plant dry weight (21.83%), shoot dry weight (26.04%) and root dry weight (8.33%) under WB (Table 2). Meanwhile, in “Yanzhi”, the inhibition effect of WB was only observed in the root dry weight (23.33%) (Table 2). Besides, the supplementary lights did not affect the root to shoot ratio of “Yanzhi” while that of “Red Butter” was markedly decreased by WFR and increased by WB (Table 2). The interaction of light quality and cultivar also confirmed that “Yanzhi” and “Red Butter” responded differently to blue light as regards plant fresh weight, shoot fresh weight (*p* < 0.001), root dry weight (*p* < 0.05) and root to shoot ratio (*p* < 0.001) (Table 2).

### 3.2. Foliage Color and Pigment Contents of Lettuce

The color characteristics (L*, a* and b*) of lettuce leaf were significantly affected by light quality, cultivars and their interaction (*p* < 0.001) (Table 3). Lettuce grown under W showed the greener leaves, while lettuce grown under WFR had the brighter and yellower leaves and lettuce grown under WB had darker and redder leaves. Not surprisingly, the pigment contents were in favor of the leaf coloration (Table 4). While both light quality and cultivar had significant effects on the accumulation of pigments, their interaction was only observed in TA contents (*p* < 0.001) (Table 4). In “Yanzhi”, the contents of Chl a, Chl b and TA under WB were 23.54%, 27.22% and 116.03% higher than under W, respectively. As compared to W, the TA content of “Yanzhi” under WFR decreased by 24.02%. In “Red Butter”, the contents of Chl a, Chl b and Chl (a + b) under WFR were 23.16%, 25.46% and 23.69% lower than those under W, respectively. The TA content of “Red Butter” under WB increased by 148.35%, compared to W, while those under WFR were unaffected. Besides, no significant differences were found in the Car contents between pairs of WB and W or WFR and W. These suggested that chlorophylls and anthocyanins might be the main factors leading to the differential coloration of lettuce, though two lettuce cultivars presented distinct sensitivity to light quality.

### 3.3. Antioxidant Sctivities of Lettuce

The cultivar differences showed that “Yanzhi” possessed better antioxidant capability (*p* < 0.001) (Table 5). However, the DPPH and FRAP of lettuce seemed to be little affected by light quality in both cultivars, except that the FRAP of “Red Butter” was increased by WB.

### 3.4. Antioxidant Compounds of Lettuce

The antioxidant compounds were regarded as the health-promoting compounds for human diets. The cultivar differences showed that “Yanzhi” possessed higher contents of TPC and VC, while “Red Butter” had richer VA content (*p* < 0.001) (Table 6). Meanwhile, light quality worked as a strong influence factor (*p* < 0.001). WB was effective in increasing the contents of TPC (22.62% and 39.46%) and TF (46.67% and 33.48%) in “Yanzhi” and “Red Butter”, compared to those under W. The TPC content of “Red Butter” was 43.05% lower while the TF content was 33.48% higher under WFR than W, respectively. The VC content of “Yanzhi” increased by 14.43% under WB, while that of “Red Butter” reduced by 23.26% by WFR (Table 6). Interestingly, the VA contents significantly accumulated under WFR (36.71% and 29.08%) and dramatically induced by WB (99.43% and 51.58%) in both “Yanzhi” and “Red Butter”.

### 3.5. Nutritional Qualities of Lettuce

The nutritional compounds of different lettuce cultivars showed significant differences that “Yanzhi” possessed higher contents of SS and SP, while “Red Butter” had higher nitrate content (*p* < 0.001) (Table 7). The light quality also influenced the accumulation of these compounds (*p* < 0.05 or 0.001). In “Yanzhi”, the contents of SP and SS were not affected by light quality, while in “Red Butter” those were 20.06% and 47.21% higher in WFR than W, respectively. The SP content of “Red Butter” decreased by 17.04% under WB, compared to W. Both in “Yanzhi” and “Red Butter”, the nitrate content significantly increased, 11.86% and 9.97% under WFR, respectively.

### 3.6. Principal Compound Analysis and Heatmap Analysis

To explore the response pattern of lettuce under different light quality and to statistically analyze the correlation among plant biomass, color parameters and phytochemicals, the principal compound analysis (Figure 2) and the heat map analysis were applied (Figure 3).

From the whole, the first four principal components (PCs) of “Yanzhi” were connected to eigenvalues higher than 1 (Figure 2b) and explained 90.53% of the cumulative variance, with PC1 accounting for 56.37% and PC2 for 20.58% (Figure 2c). In “Red Butter”, the first three PCs were associated with eigenvalues higher than 1 (Figure 2e) and explained 94.35% of the cumulative variance, with PC1 accounting for 75.85% and PC2 for 13.62% (Figure 2f). The biplots visually presented the correlations between two parameters (Figure 2a,d). For instance, positive correlations were found among Chl a, Chl b, Chl (a + b) and Car in both lettuce cultivars (Figure 2a,d). TA was positively correlated to a*, while it was negatively related to b* and L* (Figure 2a,d). However, in “Yanzhi” the color parameter a* was closely correlated to all tested pigments (Figure 2a), while in “Red Butter” the a* was little affected by pigments such as Chl a, Chl b, Chl (a + b) and Car (Figure 2b).

Additionally, the clustered heat maps well complemented and verified the results generated by PCAs. In “Yanzhi”, the light quality was divided into three main clusters (Figure 3a). The cluster WFR was characterized by higher plant biomass which contributed to the separation of WFR from the other two clusters (Figure 3a). The cluster WB was characterized by lower plant biomass and higher contents of phytochemicals (Figure 3a). Similarly, in “Red Butter”, the light quality was distinguished into three clusters (Figure 3b). Clusters W and WB were close to each other, while cluster WFR was apart from the other two clusters. WFR was defined by higher plant biomass and lower contents of phytochemicals (Figure 3b).

## 4. Discussion

Two cultivars of red-leaf lettuce were subjected to three light qualities in the PFAL; the cultivar responded differently to WFR and WB. The leaf color can be largely affected by environmental factors, especially the light conditions. Light spectra with higher percentage of blue light have been proven to positively impact on the accumulation of chlorophylls in lettuce, spinach, kale, basil pepper and cucumber leaves [31,32,33], as well as the anthocyanins synthesis in *Cryptanthus*, *Guzmania* and *Hypoestes* [34]. In contrast, far-red light was reported to decrease the contents of chlorophylls, carotenoids and anthocyanins in lettuce [16,29,35]. Similarly, in this study the WB helped to produce red foliage (a* = 2.34, b* = 7.68) while WFR contributed to pale green leaves (a* = −4.19, b* = 23.51) (Figure 1 and Table 3). This phenotype was consistent with the pigment contents in “Yanzhi” and “Red Butter” (Table 4). In the meantime, the PCA clearly showed that the anthocyanins (TA) were significantly and positively relevant to the red color (a*), while negatively correlated to the green color (b*) of lettuce leaves (Figure 2). Thus, the anthocyanins might be the predominant pigments contributing to the red color of lettuce. Two lettuce cultivars responded differently to additional far-red and blue light, the pigments content in “Yanzhi” seemed to be more sensitive to WFR, while those in “Red Butter” were more susceptible to WB.

The blue light could also enhance the photosynthesis efficiency by regulating the stomatal opening, obtaining more CO_2_ and suppressing photorespiration [36]. Consequently, additional blue light had the ability to promote the biomass of plants such as Chinese kale and tomato [37,38]. However, in this study the increased photosynthetic pigments did not improve the biomass of “Yanzhi” under WB, and the WB even decreased fresh and dry weight of “Red Butter” though its photosynthetic pigments were unchanged (Table 2). Since the effects of WB on the pigments and biomass of lettuce in this study were inconsistent with those of previous studies, further exploration such as photosynthetic characters and anatomical evidence is needed for the WB inhibited lettuce growth.

While far-red light was reported to induce shade-avoidance syndromes and reduce the plant weight [12,16], it functioned as a favorable factor for plant biomass accumulation under specific experimental conditions [18,39]. Surprisingly, in this study WFR could lead to significant higher biomass of “Yanzhi” (43.93%~51.38%) and “Red Butter” (45.11%~97.45%) (Table 2). Although far-red light is beyond the visible light, it played a pivotal role in photosynthesis [40]. The supplementary far-red light could increase the quantum yield of photosystem II and net photosynthesis, while it reduced the non-photochemical quenching of fluorescence [17,41]. As a result, the radiation use efficiency enhanced and the plant biomass increased. The shade-avoidance syndrome stimulated by far-red light, such as the promoted shoot elongation and the enlarged leaf [35,42,43,44,45], probably allowed better light interception and resulted in a remarkable increase in biomass in the PFALs. Besides, these results suggested that “Red Butter” was more sensitive to an ambient light environment than “Yanzhi” and was largely manifested in greater biomass under WFR and sharply decreased yield under WB.

Furthermore, the heat map analysis showed that the nitrates and the fresh weight and dry weight of the edible parts in both lettuce cultivars were in the same cluster under different light treatments, indicating the importance of nitrogen to plant growth (Figure 3). WFR led to significantly higher nitrate content in two cultivars (Table 7), indicating that a better absorption of N nutrient can be induced by WFR for the better growth and higher biomass of lettuce. The PCA also confirmed that a significant positive correlation existed between nitrates and biomass in two cultivars (Figure 2 and Figure 3). Besides, it should be mentioned that the nitrate contents (0.52~1.07 mg·g^−1^ FW) in the two cultivars were far below the general level (2.5~4.5 mg·g^−1^ FW) of lettuce [46]. Thus, the increased nitrates by WFR caused little harm to human health.

In general, blue light can strengthen the antioxidant capacities of vegetables through light signaling or creating light stress [47,48,49]. However, in this study neither enhancement nor reduction were found in DPPH and FRAP in two lettuce cultivars under light treatments (Table 5).

Light spectra also influence the phytochemical accumulation in plants. With respect to health-promoting compounds, TA, TPC and TF were determined in both lettuce cultivars (Table 4 and Table 6). These metabolites had potent abilities to prevent numerous chronic diseases and reduce the risk of cancers in humans [50,51]. Moreover, flavonoids and phenolics are also important for the vegetable’s flavor [52,53]. As reported before, additional blue light resulted in higher contents of anthocyanins in dropwort and Chinese kale [54,55,56], flavonols in basil, lettuce and garden rocket [57] and phenolics in pak-choi [37]. Similar results were found in this study (Table 4 and Table 6). It is known that the biosynthetic pathways of phenolics, flavonoids and anthocyanins are all derived from the phenylpropanoid biosynthesis. Blue light could regulate the expression of key enzyme genes in these metabolic pathways, such as *PAL*, *F3H*, *CHS*, *ANS* and *GST* [58,59]. Thus, it was not surprising that the TA, TPC and TF were increased by WB in “Yanzhi” and in “Red Butter” (Table 4 and Table 6).

The VC is known for its antioxidant properties. The VC functions as a redox buffer that reduces, and thereby neutralizes, reactive oxygen species [60]. No changes in ascorbic acid were found in Chinese kale sprouts treated with blue light [61], whereas VC contents in lettuce, pak-choi and Chinese kale were significantly improved by supplemental blue light [16,37,55,56]. In this study, the VC content of “Yanzhi” was improved by WB, while it was not affected in “Red Butter” (Table 6). These results indicated that the VC response to lights varied between cultivars. Another possibility is that the VC did not respond to light directly, but to the changes of other antioxidant-related processes caused by light [62].

Similarly, the VA also has nutritional and antioxidant functions. The VA protects the cholesterol from oxidation by quenching oxidants like superoxide, hydroxyl and peroxide radicals [63]. The VA in human diets usually has animal origins. As for fruits and vegetables, studies preferred to focus on VA precursors, such as carotenoids. In the greenhouse light conditions, supplementary blue light significantly increased the lycopene content in tomato fruits from 42 to 54 days after anthesis, as well as the β-carotene content from 36 to 54 days after anthesis [64]. The supplementary red light could also increase the contents of phytoene and lycopene of tomato fruits at breaker stage [65]. In this study, both VA and Car were determined (Table 4 and Table 6). However, the Car content seemed to be little affected by WFR or WB in two lettuce cultivars (Table 4), indicating the species differences of carotenoids in response to light qualities. Surprisingly, the VA contents were significantly increased by WFR and WB in both lettuce cultivars (Table 6). Thus, this study speculated that WFR and WB might inhibit VA degradation by down-regulation of genes or suppression of the enzyme activities related to VA metabolism and consequently VA contents accumulated in lettuce. While the regulation mechanisms require further investigation through molecule methods, these results still suggested that supplementary far-red and blue lights could be used as effective tools to elevate the lettuce quality.

In addition, the nutritional compounds such as SP and SS of two lettuce cultivars seemed to be less sensitive to WB, while these were significantly increased by WFR in both cultivars (Table 7). These results suggested that the effects of WB and WFR on phytochemicals were vegetable species and cultivar dependent.

## 5. Conclusions

In PFALs, under 250 μmol·m^−2^·s^−1^ PPFD LED, the supplementary blue light could be applied to deepen the red color of lettuce leaves via accumulating the TA contents. Meanwhile, it benefited the flavor formation of lettuce by increasing the contents of TPC, TF and SS. Besides, the supplementary blue light drastically increased the VA content in two lettuce cultivars and the VC content in lettuce “Yanzhi”, which indicated that blue LEDs could be used to elevate the lettuce quality by making richer its health-promoting compounds. As for supplementary far-red light, it obviously favored the biomass of two lettuce cultivars through higher N uptake, which was of great significance to increase the vegetable production. In addition, although the far-red light was not as superior as blue light in improving crop qualities, it still had the ability to increase the contents of SP and SS in lettuce “Red Butter” and the VA content in the two lettuce cultivars. Thus, this study suggested that both far-red and blue LEDs have promising application prospects in PFALs as supplemental lights.

## Figures and Tables

**Figure 1 molecules-26-07405-f001:**
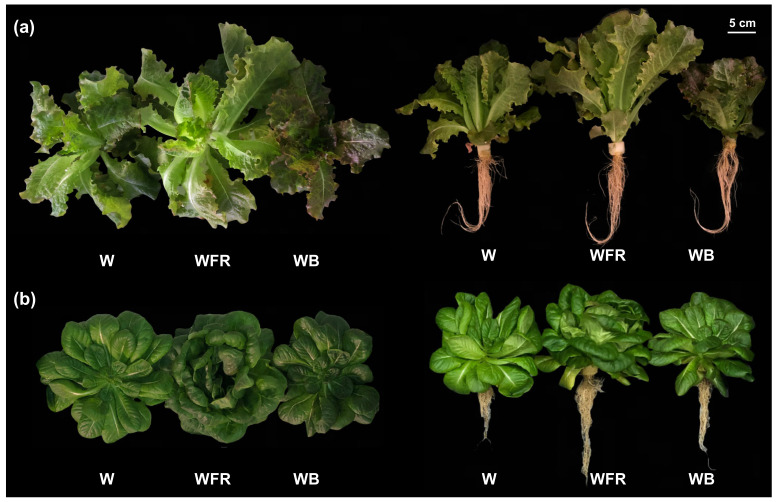
The morphology of two cultivars of red-leaf lettuce under different light quality. The lettuce (**a**) “Yanzhi” and (**b**) “Red Butter”. W = equalized-white light; WFR = W plus far-red light; WB = W plus blue light.

**Figure 2 molecules-26-07405-f002:**
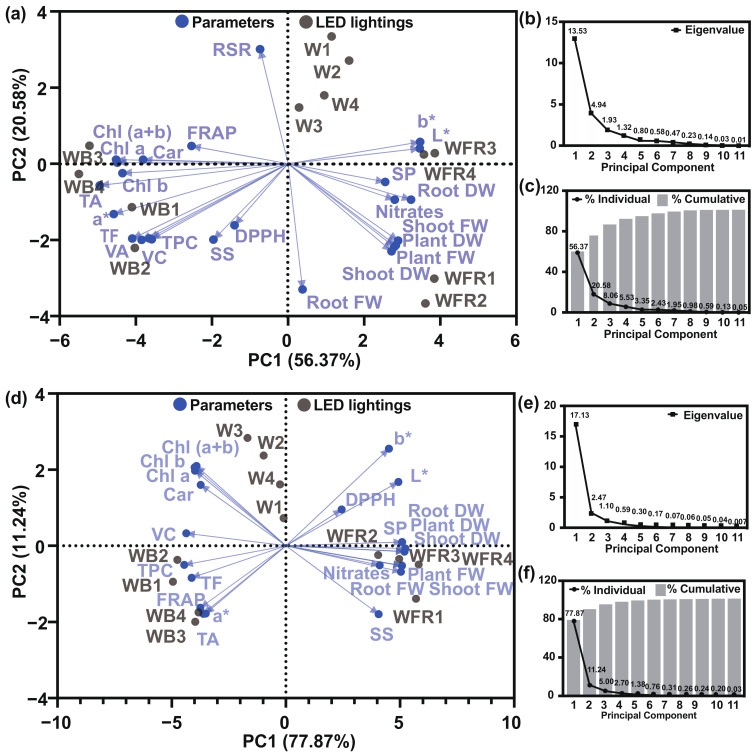
The principal compound analysis of overall quality of two lettuce cultivars. The biplot (**a**), eigenvalue (**b**) and variance proportion (**c**) of “Yanzhi”. The biplot (**d**), eigenvalue (**e**) and variance proportion (**f**) of “Red Butter”. FW = fresh weight, DW = dry weight, L* = lightness, a* = redness, b* = yellowness, Chl = chlorophyll, Car = carotenoids, TA = total anthocyanins, DPPH = DPPH radical inhibition percentage, FRAP = ferric ion reducing antioxidant power, TF = total flavonoids, TPC = total phenolic compounds, SP = soluble proteins, SS = soluble sugars, RSR = root to shoot ratio, W = equalized-white light; WFR = W plus far-red light; WB = W plus blue light.

**Figure 3 molecules-26-07405-f003:**
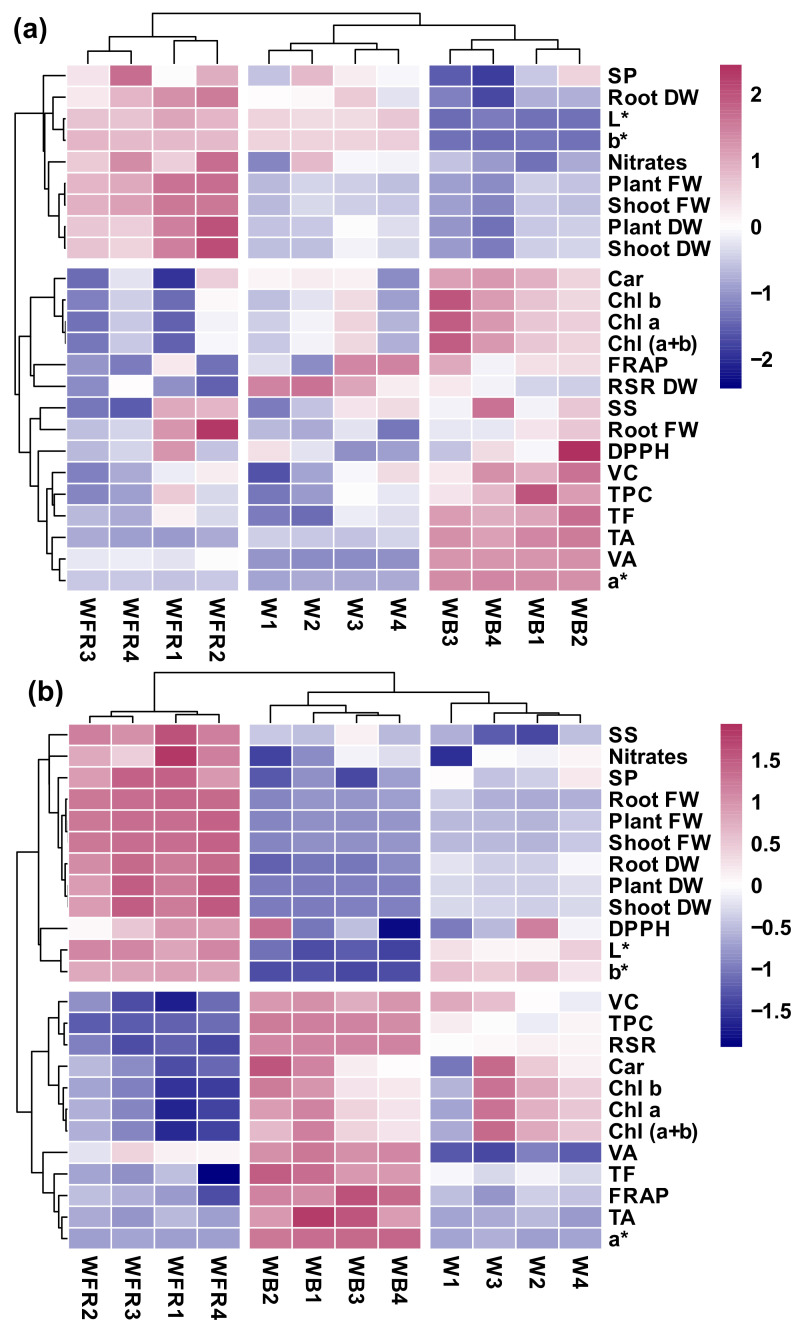
The heat map analysis of overall quality of two lettuce cultivars. The clustered heat map of (**a**) “Yanzhi” and (**b**) “Red Butter”. FW = fresh weight, DW = dry weight, L* = lightness, a* = redness, b* = yellowness, Chl = chlorophyll, Car = carotenoids, TA = total anthocyanins, DPPH = DPPH radical inhibition percentage, FRAP = ferric ion reducing antioxidant power, TF = total flavonoids, TPC = total phenolic compounds, SP = soluble proteins, SS = soluble sugars, RSR = root to shoot ratio, W = equalized-white light; WFR = W plus far-red light; WB = W plus blue light.

**Table 1 molecules-26-07405-t001:** Spectral data for the light-emitting diodes (LEDs).

Parameters	Lightings ^1^		
	W	WFR	WB
Single-band photon flux density (μmol·m^−2^·s^−1^)			
Blue light (400–500 nm)	31.41	29.66	72.25
Green light (500–600 nm)	47.92	46.51	36.34
Red light (600–700 nm)	170.73	168.96	137.22
Far-red light (700–800 nm)	3.30	13.24	3.59
Integrated photon flux density ^2^ (μmol·m^−2^·s^−1^)			
PPFD (400–700 nm)	249.06	245.13	245.80
YPFD (400–700 nm)	223.73	219.76	212.42
Radiation ratio			
Red/Blue	5.44	5.70	1.90
Red/Green	3.56	3.63	3.78
Red/Far-red	51.70	12.76	38.25
Blue/Far-red	9.51	2.24	20.14
Daily light integral (mol·m^−2^·day^−1^)			
10 h	9.00	8.82	8.85

^1^ W = equalized-white light; WFR = W plus far-red light; WB = W plus blue light. ^2^ PPFD: The photosynthetic photon flux density; YPFD: The yield photon flux density.

**Table 2 molecules-26-07405-t002:** Effects of different light quality on plant biomass of lettuce.

	Fresh Weight (g)			Dry Weight (g)			Root/Shoot
	Plant	Shoot	Root	Plant	Shoot	Root	
Yanzhi							
W ^1^	53.41 ± 1.66b ^2^	47.99 ± 1.55b	5.42 ± 0.44a	2.68 ± 0.19b	2.08 ± 0.17b	0.60 ± 0.04a	0.29 ± 0.02a
WFR	77.93 ± 5.82a	71.07 ± 4.45a	6.86 ± 1.44a	3.85 ± 0.53a	3.15 ± 0.47a	0.71 ± 0.07a	0.23 ± 0.02b
WB	50.64 ± 4.15b	44.30 ± 3.77b	6.35 ± 0.44a	2.32 ± 0.32b	1.85 ± 0.26b	0.46 ± 0.05b	0.25 ± 0.01b
Red Butter							
W	50.65 ± 1.81b	46.44 ± 1.82b	4.22 ± 0.12b	2.52 ± 0.05b	1.92 ± 0.05b	0.60 ± 0.01b	0.31 ± 0.01b
WFR	97.81 ± 2.00a	91.69 ± 1.96a	6.12 ± 0.06a	3.87 ± 0.24a	3.18 ± 0.23a	0.69 ± 0.01a	0.22 ± 0.01c
WB	41.96 ± 1.09c	38.00 ± 1.00c	3.96 ± 0.09c	1.97 ± 0.03c	1.42 ± 0.02c	0.55 ± 0.01c	0.39 ± 0.00a
Interaction ^3^							
Light quality (L)	***	***	***	***	***	***	***
Cultivar (C)	*	***	***	NS	NS	NS	***
L × C	***	***	NS	NS	NS	*	***

^1^ W = equalized-white light; WFR = W plus far-red light; WB = W plus blue light. ^2^ Data are expressed as mean ± standard deviations (SD). Lowercase letters within columns indicate significant differences at *p* < 0.05 according to one-way ANOVA. ^3^ NS, * and *** represent non-significant or significant at *p* < 0.05 and 0.001, respectively, according to two-way ANOVA, Tukey’s honest significant difference tests.

**Table 3 molecules-26-07405-t003:** Effects of different light quality on color parameters of lettuce.

	L* ^1^	a*	b*
Yanzhi			
W ^2^	39.18 ± 0.56b ^3^	−5.26 ± 0.03c	21.48 ± 0.21b
WFR	40.89 ± 0.78a	−4.19 ± 0.10b	23.51 ± 0.19a
WB	29.75 ± 0.29c	2.34 ± 0.15a	7.68 ± 0.24c
Red Butter			
W	41.06 ± 0.51b	−10.28 ± 0.09b	27.73 ± 0.80b
WFR	43.45 ± 0.37a	−10.35 ± 0.04b	29.33 ± 0.10a
WB	36.71 ± 0.46c	−6.75 ± 0.11a	19.10 ± 0.14c
Interaction ^4^			
Light quality (L)	***	***	***
Cultivar (C)	***	***	***
L × C	***	***	***

^1^ L* = lightness, a* = redness, b* = yellowness. ^2^ W = equalized-white light; WFR = W plus far-red light; WB = W plus blue light. ^3^ Data are expressed as mean ± standard deviations (SD). Lowercase letters within columns indicate significant differences at *p* < 0.05 according to one-way ANOVA. ^4^ *** represents significant at *p* < 0.001, according to two-way ANOVA, Tukey’s honest significant difference tests.

**Table 4 molecules-26-07405-t004:** Effects of different light quality on pigment contents of lettuce.

	Chlorophylls (mg·g^−1^)			Carotenoids (mg·g^−1^)	TA (mg·g^−1^)
	Chl a ^1^	Chl b	Chl (a + b)		
Yanzhi					
W ^2^	0.84 ± 0.08b ^3^	0.26 ± 0.03b	1.11 ± 0.11a	0.19 ± 0.01ab	6.72 ± 0.24b
WFR	0.73 ± 0.10b	0.24 ± 0.03b	0.98 ± 0.14a	0.18 ± 0.02b	5.11 ± 0.32c
WB	1.04 ± 0.09a	0.33 ± 0.03a	1.38 ± 0.13a	0.21 ± 0.01a	14.53 ± 0.73a
Red Butter					
W	1.26 ± 0.15a	0.38 ± 0.05a	1.66 ± 0.20a	0.42 ± 0.09ab	0.24 ± 0.02b
WFR	0.97 ± 0.09b	0.29 ± 0.02b	1.27 ± 0.11b	0.32 ± 0.03b	0.24 ± 0.02b
WB	1.30 ± 0.08a	0.40 ± 0.03a	1.69 ± 0.10a	0.47 ± 0.06a	0.60 ± 0.08a
Interaction ^4^					
Light quality (L)	***	***	***	**	***
Cultivar (C)	***	***	***	***	***
L × C	NS	NS	NS	NS	***

^1^ Chl = chlorophyll; TA = total anthocyanins. ^2^ W = equalized-white light; WFR = W plus far-red light; WB = W plus blue light. ^3^ Data are expressed as mean ± standard deviations (SD). Lowercase letters within columns indicate significant differences at *p* < 0.05 according to one-way ANOVA. ^4^ NS, ** and *** represent non-significant or significant at *p* < 0.01 and 0.001, respectively, according to two-way ANOVA, Tukey’s honest significant difference tests.

**Table 5 molecules-26-07405-t005:** Effects of different light quality on antioxidant activities of lettuce.

	DPPH ^1^ (%)	FRAP (mmol·g^−1^)
Yanzhi		
W ^2^	91.92 ± 0.62a ^3^	0.02 ± 0.00a
WFR	92.29 ± 0.87a	0.02 ± 0.00a
WB	92.91 ± 1.28a	0.02 ± 0.00a
Red Butter		
W	90.49 ± 0.91a	0.01 ± 0.00b
WFR	91.19 ± 0.40a	0.01 ± 0.00b
WB	90.10 ± 1.31a	0.02 ± 0.00a
Interaction ^4^		
Light quality (L)	NS	***
Cultivar (C)	***	***
L × C	NS	***

^1^ DPPH = DPPH radical inhibition percentage; FRAP = ferric ion reducing antioxidant power. ^2^ W = equalized-white light; WFR = W plus far-red light; WB = W plus blue light. ^3^ Data are expressed as mean ± standard deviations (SD). Lowercase letters within columns indicate significant differences at *p* < 0.05 according to one-way ANOVA. ^4^ NS, *** represent non-significant or significant at *p* < 0.001, according to two-way ANOVA, Tukey’s honest significant difference tests.

**Table 6 molecules-26-07405-t006:** Effects of different light quality on antioxidant compounds of lettuce.

	TPC^1^ (mgGAE·g^−1^)	TF (mg·g^−1^)	VC (mg·g^−1^)	VA (nmol·g^−1^)
Yanzhi				
W ^2^	1.07 ± 0.09b ^3^	1.76 ± 0.26b	0.83 ± 0.07b	5.23 ± 0.14c ^2^
WFR	1.10 ± 0.11b	1.91 ± 0.18b	0.84 ± 0.05b	7.15 ± 0.29b
WB	1.32 ± 0.10a	2.58 ± 0.13a	0.95 ± 0.05a	10.43 ± 0.06a
Red Butter				
W	0.68 ± 0.03b	2.10 ± 0.06b	0.43 ± 0.03a	7.29 ± 0.34c
WFR	0.39 ± 0.01c	1.69 ± 0.34b	0.33 ± 0.02b	9.41 ± 0.51b
WB	0.95 ± 0.02a	2.81 ± 0.15a	0.47 ± 0.01a	11.05 ± 0.23a
Interaction ^4^				
Light quality (L)	***	***	***	***
Cultivar (C)	***	NS	***	***
L × C	***	*	NS	***

^1^ TPC = total phenolic compounds; TF = total flavonoids; VC = vitamin C; VA = vitamin A. ^2^ W = equalized-white light; WFR = W plus far-red light; WB = W plus blue light. ^3^ Data are expressed as mean ± standard deviations (SD). Lowercase letters within columns indicate significant differences at *p* < 0.05 according to one-way ANOVA. ^4^ NS, * and *** represent non-significant or significant at *p* < 0.05 and 0.001, respectively, according to two-way ANOVA, Tukey’s honest significant difference tests.

**Table 7 molecules-26-07405-t007:** Effects of different light quality on nutritional compound contents in lettuce.

	Soluble Proteins (mg·g^−1^)	Soluble Sugars (mg·g^−1^)	Nitrates (mg·g^−1^)
Yanzhi			
W ^1^	11.50 ± 0.50a	30.33 ± 3.39a	0.57 ± 0.05b
WFR	12.04 ± 0.66a	30.41 ± 5.90a	0.65 ± 0.04a
WB	10.69 ± 0.92a	33.63 ± 3.66a	0.52 ± 0.02b
Red Butter			
W	7.46 ± 0.43b	12.47 ± 2.30b	0.97 ± 0.06b
WFR	9.34 ± 0.42a	23.63 ± 1.34a	1.07 ± 0.04a
WB	6.19 ± 0.47c	15.57 ± 1.58b	0.95 ± 0.04b
Interaction ^3^			
Light quality (L)	***	*	***
Cultivar (C)	***	***	***
L × C	*	**	NS

^1^ W = equalized-white light; WFR = W plus far-red light; WB = W plus blue light. ^2^ Data are expressed as mean ± standard deviations (SD). Lowercase letters within columns indicate significant differences at *p* < 0.05 according to one-way ANOVA. ^3^ NS, *, **, *** represent non-significant or significant at *p* < 0.05, 0.01, and 0.001, respectively, according to two-way ANOVA, Tukey’s honest significant difference tests.

## Data Availability

The datasets generated and/or analyzed during the study can be obtained from the corresponding author on request.

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
