# Peer review of "Supplementary Far-Red and Blue Lights Influence the Biomass and Phytochemical Profiles of Two Lettuce Cultivars in Plant Factory"

_molecules, 2021, doi:10.3390/molecules26237405_

Round 1

Reviewer 1 Report

I would like to inform you that this paper is acceptable at this time. The manuscript has been well arranged.

Minor point.

*Some discussion sentences are included in the part of the results. Please move it to the discussion section.

*Match the format of tables. Some tables are different.

Reviewer 2 Report

Article:

molecules-1453177-Supplementary far-red and blue lights influence the biomass and phytochemical profiles of two lettuce cultivars

Even though manuscript is well written I couldn’t find any molecular aspects that warrant publication of this manuscript in an esteemed journal like MOLECULES. I am sure, this manuscript will contribute important piece of information for increasing productivity of this vegetable crop lettuce exposed to different light-intensity and radiation using LED in a modern agriculture setup – in journals like HORTICULTURAE, PLANTS, CROPS (examples from MDPI).

Before submitting to any of the above-mentioned journals, I have some specific suggestions as given below, moreover, the authors should do some additional works in order to qualify for publication. In my opinion, there should be a comparison with conventional greenhouse grown plants. They should analyze Vitamin A, as this is a leafy vegetable crop. I think, these parameters should be included in re-submission.

Title:

The tile should be self-explanatory, it should give the readers the full information. Eg: there’s no mention about the farming system (were it is done, in green house/hydroponic/vertical smart system or what??)

Abstract

There should be a conclusion statement at the end of the abstract.

Keywords

Include “Vitamins”

Introduction

  • The authors should point out the importance of PAR (photosynthetically active radiation) in the introduction section and should correlate how the provided LEDs contribute to achieve this PAR in plants (this is really important).
  • In introduction section, authors should elaborate the earlier studies done in lettuce in this system (hydro-aqua ponics) and the expected benefits over them with the current system using by authors
  • Heatmap analysis should be mentioned in objectives

Materials and Methods

  • The variety of lettuce used should be explained in detail
  • There should be detailed Experimental Setup (as section 2.2.) explaining the layout, whether it is a vertical setup, or horizontal PVC tube setup, or what. How many plants in each layer/tube, how much water in each layer etc.
  • Antioxidant components measurements should not be in one subsection, in my view, ANTIOXIDANT COMPONENTS in the article are TPC and TF. These should be in one section, ANTIOXIDANT ACTIVITY like DPPH and FRAP should be in separate section. Don’t confuse the readers with mixing of these two things.
  • Detailed methodology of “heatmap analysis” should be added as section in Materials and Methods (before Statistics)

Results

  • What is the control in this experiment? You compare with greenhouse plants? Or white light is considered as control? If yes, based on what understanding? How you decided plants in nature are receiving only white light? What about PAR in natural sunlight?
  • Figure 1 legend is not self-explanatory, what is ‘W’ ‘WFR’ ‘WB’ abbreviations should be explained in legend.

Discussion

  • Antioxidant compound contents and antioxidant activities should be separately explained in all sections throughout the manuscript.

Conclusion

  • Should not generalize

Round 2

Reviewer 2 Report

As the authors considered all suggestions and the editors confirmed the suitability of the article in MOLECULES  i recommends acceptance.